# Neural correlates of obesity across the lifespan
Filip Morys [1] ✉, Christina Tremblay[1], Shady Rahayel[2,3], Justine Y. Hansen [1], Alyssa Dai[1], Bratislav Misic [1] & Alain Dagher [1]

Associations between brain and obesity are bidirectional: changes in brain structure and function underpin over-eating, while chronic adiposity leads to brain atrophy. Investigating brain-obesity interactions across the lifespan can help better understand these relationships. This study explores the interaction between obesity and cortical morphometry in children, young adults, adults, and older adults. We also investigate the genetic, neurochemical, and cognitive correlates of the brain-obesity associations. Our findings reveal a pattern of lower cortical thickness in fronto-temporal brain regions associated with obesity across all age cohorts and varying age-dependent patterns in the remaining brain regions. In adults and older adults, obesity correlates with neurochemical changes and expression of inflammatory and mitochondrial genes. In children and older adults, adiposity is associated with modifications in brain regions involved in emotional and attentional processes. Thus, obesity might originate from cognitive changes during early adolescence, leading to neurodegeneration in later life through mitochondrial and inflammatory mechanisms.

The multifaceted nature of obesity, driven by an association between genetic and environmental factors[1–3], has recently become a focus of scientific research. Obesity is highly heritable and genes related to excess weight gain are predominantly expressed in the central nervous system[4]. Given that obesity is increasingly recognised as a disorder marked by brain changes that affects individuals across the lifespan, from children to older adults[5–11], understanding its neural correlates may provide critical insights into targeted interventions and preventive strategies tailored to specific age groups.

Increasing evidence suggests that chronic obesity can have significant effects on brain structure, function, connectivity, and thus cognition[12,13]. Mechanisms involved are constituents of the metabolic syndrome, namely type 2 diabetes, hypertension, inflammation, and dyslipidemia, likely exerting their effects in part via cerebrovascular disease[14]. Simultaneously, certain brain phenotypes are thought to constitute vulnerability factors for excess weight gain, possibly through behaviours such as impulsivity, uncontrolled eating, or executive dysfunction[5,15,16]. Because obesity-related genes affect neurodevelopment, and should therefore be present in early life, while degenerative, inflammatory or vascular brain changes accrue over time in individuals with excess weight, the neural correlates of obesity are likely to differ throughout the lifespan.

Brain map repositories allow researchers to gain a better understanding of neural changes associated with conditions like obesity. The Allen Human Brain Atlas (AHBA), a rich repository of microarray gene expression in the human brain, presents a unique opportunity to investigate the genetic undercurrents of obesity[17,18]. It allows for identification of cellular and molecular processes that underpin some brain changes[19–21]. Furthermore, recent integration of multiple positron emission tomography (PET) datasets by Hansen and colleagues[22] enables us to deepen our understanding of neurochemical systems potentially involved in brain changes related to excess weight. Finally, automated meta-analytic tools such as Neurosynth allow for identification of brain activation patterns related to different cognitive processes, thus enabling an indirect investigation of altered cognition in obesity[23].

Building upon this foundation, we aim to provide a comprehensive account of the interplay between brain structure, genetic factors, neurocognitive phenotypes, and obesity across the lifespan. Integrating multimodal data, including neuroimaging, neurochemistry, gene expression, and cognition across multiple age groups, will help uncover mechanisms related to obesity-brain interactions. The strategy here involves (1) using large-scale datasets (total $n > 45,000$) from the Adolescent Brain Cognitive Development (ABCD), Human Connectome Project (HCP; young adults), Human Connectome Project - Aging (HCP-A), and the UK Biobank (UKBB; older adults) to investigate cortical thickness changes related to obesity at different ages; (2) finding commonalities between those datasets and AHBA/PET/Neurosynth data on gene expression/neurotransmitter receptor and transporter distribution/cognitive processes in the brain. We focus on

[1]Montreal Neurological Institute, McGill University, H3A 2B4 Montreal, QC, Canada. [2]Department of Medicine and Medical Specialties, University of Montreal, Montreal, QC, Canada. [3]Center for Advanced Research in Sleep Medicine, Hopital du Sacre-Coeur de Montreal, Montreal, QC, Canada. ✉e-mail: filip.morys@mcgill.ca

different age groups to explore whether obesity-related brain changes have different significance and neurocognitive underpinnings across the lifespan, as previously suggested[5].

## Materials and methods

### Participants

**Adolescent Brain Cognitive Development sample.** To investigate brain correlates of obesity in early adolescence, we used the ABCD sample (data release 4.0) - a longitudinal, multi-site study[24–26]. Study procedures were approved by review boards of all participating institutions and written parental informed consent and child assent were collected for all participants. All ethical regulations relevant to human research participants were followed. We excluded all related participants, individuals with outlying BMI values (below 10 and above $50 \, kg/m^2$), and included only individuals with complete cortical thickness data. The final sample consisted of 9521 children (mean age=10 years, SD = 0.5; mean BMI = $18.92 \, kg/m^2$, SD = 4.22; 4543 girls, Fig. S1). BMI values were converted to standard deviation scores (BMI SDS) for further analysis using Center for Disease Control and Prevention's growth charts based on individual age and sex (mean BMI SDS = 0.45, SD = 1.18).

**Human Connectome Project.** To determine cortical thickness correlates of obesity in young adults, we used a sample from the HCP - a study conducted at Washington University[27]. We excluded one of each pair of twins and individuals with endocrine disorders[10]. Study procedures were approved by the relevant review board and all participants signed written informed consent forms. All ethical regulations relevant to human research participants were followed. The final sample included in our study comprised 814 participants (mean age=29 years, SD = 4; mean BMI = $26.60 \, kg/m^2$, SD = 5.28; 421 women, Fig. S1).

**Human Connectome Project – Aging.** We used a sample of the Human Connectome Project - Aging (HCP-A) to determine brain correlates of obesity in a group of middle-aged individuals. As an extension of the HCP project, the HCP-A project recruited participants between the age of 36 and 100 years at 4 sites in the USA[28,29]. Study procedures were approved by the relevant review boards and all participants gave written informed consent. All ethical regulations relevant to human research participants were followed. To minimise the overlap with the age distribution of the UK Biobank sample below, we only included individuals aged between 36 and 50 years. The final sample size was 228 (mean age = 43 years, SD = 4; mean BMI = $27.71 \, kg/m^2$, SD = 5.15; 130 women, Fig. S1).

**UK Biobank.** To derive correlations between cortical thickness and obesity measures in older adults, we used the UKBB sample - a large-scale, multi site study from the UK[30,31]. The current study was performed under UKBB study ID 45551. Prior to all analyses, we excluded individuals with neurological disorders and we only included individuals with full neuroimaging data. All participants gave written informed consent and the study was approved by the North-West Multi-Centre Research Ethics Committee. All ethical regulations relevant to human research participants were followed. The final size for this sample was 36,333 individuals (mean age = 64 years, SD = 8; mean BMI = $26.28 \, kg/m^2$, SD = 4.23; 18,696 women, Fig. S1).

### Neuroimaging data

**Adolescent Brain Cognitive Development.** Neuroimaging data were collected using 3T Siemens, General Electric, or Philips magnetic resonance imaging (MRI) scanners at 22 different sites in the United States. Data collection was harmonised by using standardised hardware and adjusting scanning sequences. Detailed imaging protocols can be found in prior publications[24]. Structural T1-weighted images with $1mm^3$ isotropic voxel size were used to obtain cortical morphometry measures. Here, we used cortical thickness data processed and provided by the ABCD initiative[32]. Data for each parcel of the Desikan-Killiany (DK; 68 parcels) atlas[33] were processed using FreeSurfer 5.3.0[34] after correcting for gradient non-linearity distortions. For this study, we used parcels corresponding to the ones from the Desikan-Killiany-Tourville (DKT; 62 parcels) atlas[35], thus omitting 3 parcels for each hemisphere for consistency with other data samples. Post-processing quality control was provided by the ABCD initiative in the form of fail/pass ratings. In the final sample, we excluded individuals who failed this step.

**Human Connectome Project.** Neuroimaging data were collected using a 3T Connectome Siemens Skyra MRI scanner. Protocol details can be found in previous publications[27]. Here, we utilised structural T1-weighted images with $0.7mm^3$ isotropic voxel size (https://www.humanconnectome.org/hcp-protocols-ya-3t-imaging). We used minimally processed data provided by the HCP[36] after gradient non-linearity distortion correction. We obtained cortical thickness data for each parcel of the DKT atlas after running FreeSurfer's 6.0.1 -*recon_all* function. We then visually inspected cortical segmentation and excluded all participants with unsatisfactory results.

**Human Connectome Project – Aging.** Neuroimaging data were collected using 3 T Siemens Prisma scanners at 4 sites using harmonised data acquisition protocols. Imaging protocols are described elsewhere[29]. Here, we utilised structural T1-weighted images with $0.8mm^3$ isotropic voxel size. We used cortical thickness data processed with FreeSurfer 6.0.0 as provided by the HCP initiative. Quality control was conducted manually to ensure proper quality of cortical segmentation. Individuals who did not pass quality control were excluded from the final sample. Finally, cortical thickness data were parcellated using the DKT atlas.

**UK Biobank.** Neuroimaging data were collected using 3T Siemens Skyra scanners at 3 sites in the UK. Imaging protocol details are described online at https://biobank.ctsu.ox.ac.uk/crystal/refer.cgi?id=2367 and in previous publications[31]. We used T1-weighted structural images with a $0.8mm^3$ isotropic voxel size. Here, we used cortical thickness imaging-derived phenotypes for each parcel of the DKT atlas provided by the UKBB[37,38]. Data were obtained using FreeSurfer 6.0.0 and quality control was performed by the UKBB initiative.

### Positron emission tomography data

To analyse spatial correspondence between obesity maps and PET neurotransmitter receptor/transporter maps, we used curated data from Hansen et al. [22], as implemented in the 'neuromaps' package[39]. 'Neuromaps' is a toolbox that contains several curated brain maps and software tools that allow researchers to make comparisons between different brain maps. The dataset used here contains atlas data from over 1,200 individuals of 19 neurotransmitter systems (receptors and transporters): dopamine $D_1$ receptor[40], $D_2$ receptor[41–44], dopamine transporter[45,46], serotonin 1a receptor[47,48], serotonin 1b receptor[47–49], serotonin 2a receptor[47,48], serotonin 4 receptor[48], serotonin 6 receptor[50], serotonin transporter[47,48], noradrenaline transporter[51,52], α4β2 nicotinic receptor[53], cannabinoid receptor 1[54,55], $GABA_a$ receptor[45,56], histamine H3 receptor[57], muscarinic M1 receptor[58], metabotropic glutamate receptor 5[59,60], μ-opioid receptor[61,62], and vesicular acetylcholine transporter[63,64]. For a detailed list of studies from which the data are derived see Supplementary Data 3 in ref. 22. Original data were parcellated using the DKT atlas for further processing using 'neuromaps'[39].

### Allen Human Brain Atlas dataset

To explore the relationship between gene expression in brain tissue and obesity maps, we used gene expression data from the AHBA[18]. Briefly, this dataset contains microarray gene expression data from 6 post-mortem brains. The average age of brain donors is 43 years (range: 24–57 years). Analysis of this dataset was restricted to the left hemisphere, as the right hemisphere data are available from only 2 post-mortem brains. We used the 'abagen' toolbox for preprocessing of the genetic data according to previous

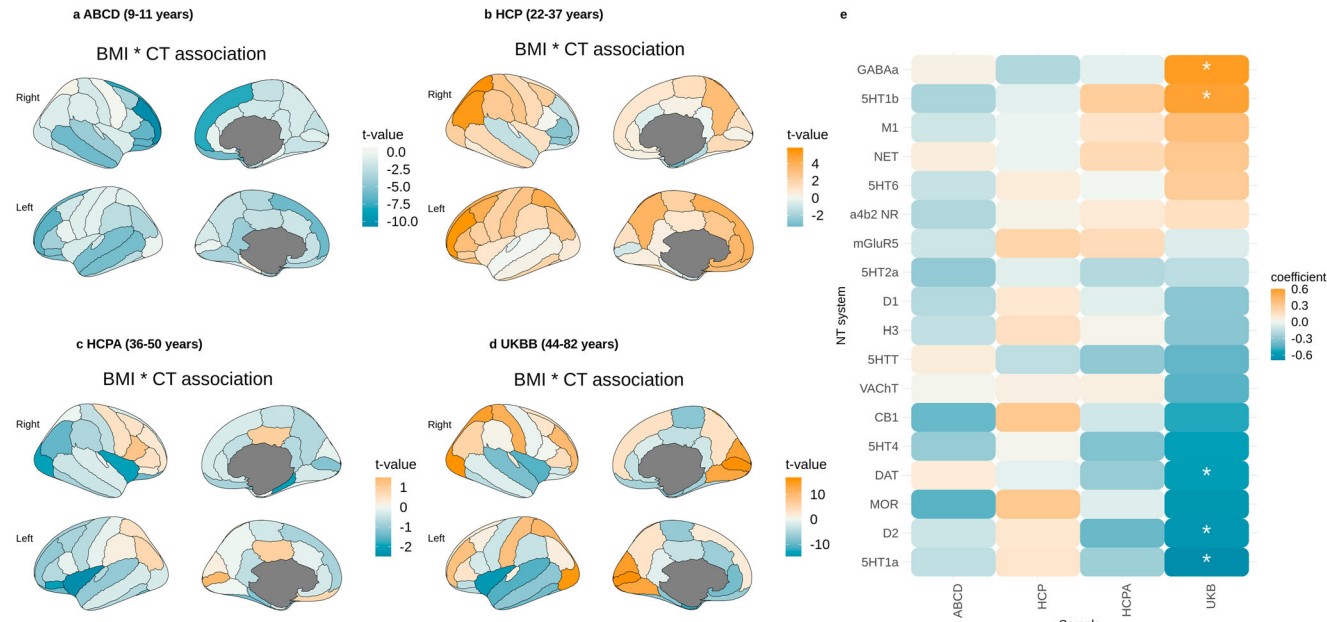

**Fig. 1 | Obesity maps and their relationship with neurotransmitter systems.**
**a** Relationship between BMI and cortical thickness in the ABCD sample;
**b** Relationship between BMI and cortical thickness in the HCP sample;
**c** Relationship between BMI and cortical thickness in the HCP-A sample;
**d** Relationship between BMI and cortical thickness in the UKBB sample;
**e** correlations between obesity maps and neurotransmitter systems (chosen neurotransmitter maps derived from largest samples - see Fig. S3 for all maps); * denotes significant associations. BMI body mass index, CT cortical thickness, GABAa

gamma-aminobutyric acid receptor a. 5HT1b - serotonin 1b receptor. M1 - muscarinic M1 receptor. NET - norepinephrine transporter. 5HT6 - serotonin 6 receptor. a4b2 NR - α4β2 nicotinic receptor. mGluR5 - metabotropic glutamate receptor 5. 5HT2a - serotonin 2a receptor. D1 - dopamine D1 receptor. H3 - histamine H3 receptor. 5HTT - serotonin transporter. VAChT - vesicular acetylcholine transporter. CB1 - cannabinoid receptor 1. 5HT4 - serotonin 4 receptor. DAT - dopamine transporter. MOR - μ-opioid receptor. D2 - dopamine D2 receptor. 5HT1a - serotonin 1a receptor.

recommendations[65,66], which resulted in expression maps of 15,633 genes. We mapped the data to the DKT parcellation for further analyses.

**BrainSpan dataset**
Because the average age of AHBA donors is 43 years and gene expression in the brain is different in childhood[67], we replicated our analysis comparing brain gene expression and obesity maps from the ABCD dataset using a sample of donors from the BrainSpan dataset (www. brainspan.org)[68]. We selected 5 specimens obtained from children around the age of the ABCD dataset (mean age: 10 years; range: 4-15 years) and analysed gene expression of 13,786 genes. To enable comparison between datasets, we only selected genes whose expression data were also available in the AHBA atlas. In the BrainSpan dataset, only 11 cortical regions of interest are available. We mapped these regions onto 11 DKT parcels as in ref. 69 for further analyses.

**Neurosynth cognitive processes dataset**
Next, we investigated how brain changes in obesity were associated with cognitive function. To this end, we used data from Neurosynth, a meta-analytic database of over 15,000 functional MRI studies (www.neurosynth. org)[23]. Every brain voxel regional value extracted from the database represents the probability of said voxel being activated in a task corresponding to a specific meta-analytic term. As previously[70], we used 123 cognitive process terms from the Cognitive Atlas (www.cognitiveatlas.org)[71]. Maps of all cognitive processes can be found at https://github.com/netneurolab/ hansen_receptors. A list of included cognitive processes can be found in the supplement to ref. 22. Data were parcellated according to the DKT atlas for further analyses.

**Statistics and reproducibility**
Scripts associated with this study can be found at https://github.com/ FilipMorys/Obesity_maps and in ref. 72.

**Associations between BMI and cortical thickness.** We used linear regression analyses to investigate the association between each parcel of the DKT atlas and BMI, separately for each sample. In the analysis, we corrected for age, sex, imaging site (for multisite studies), education (parental education in the ABCD sample), socioeconomic status (household income in the ABCD sample, individual income in the HCP and HCP-A samples, Townsend deprivation index in the UKBB sample[73]), and additionally for scanning date in the UKBB, as per previous recommendations[74]. We also repeated the analyses without regressing out the socioeconomic status indicators. Obesity maps were created as t-values of the associations between cortical thickness and BMI for each parcel of the DKT atlas.

**Neurotransmitter-cortical thickness spatial correspondence analysis.** Correspondence analysis between obesity maps and neurotransmitter maps were performed on the DKT parcellated data using 'neuromaps'. We used Pearson correlation analysis to determine correlation coefficients, and permutation spin tests (n = 10,000) for parcellated data to determine p-values while accounting for spatial autocorrelation in brain data[75,76]. Spin tests project data onto a sphere and rotate it to create spatially constrained permuted datasets for significance testing[75]. Correlation coefficients are recalculated for each of the permuted datasets thus creating a null distribution against which the true correlation coefficient is compared. Here, the significance threshold α was 0.05, meaning that the correlation coefficient was deemed significant if its absolute value was higher than 95% of the coefficients in the null distribution. Analyses were corrected for multiple comparisons using false discovery rate correction within each sample[77].

**Gene-cortical thickness spatial correspondence analysis.** We used partial least squares (PLS) regression analysis to investigate the correspondence between gene expression and obesity maps. PLS finds latent

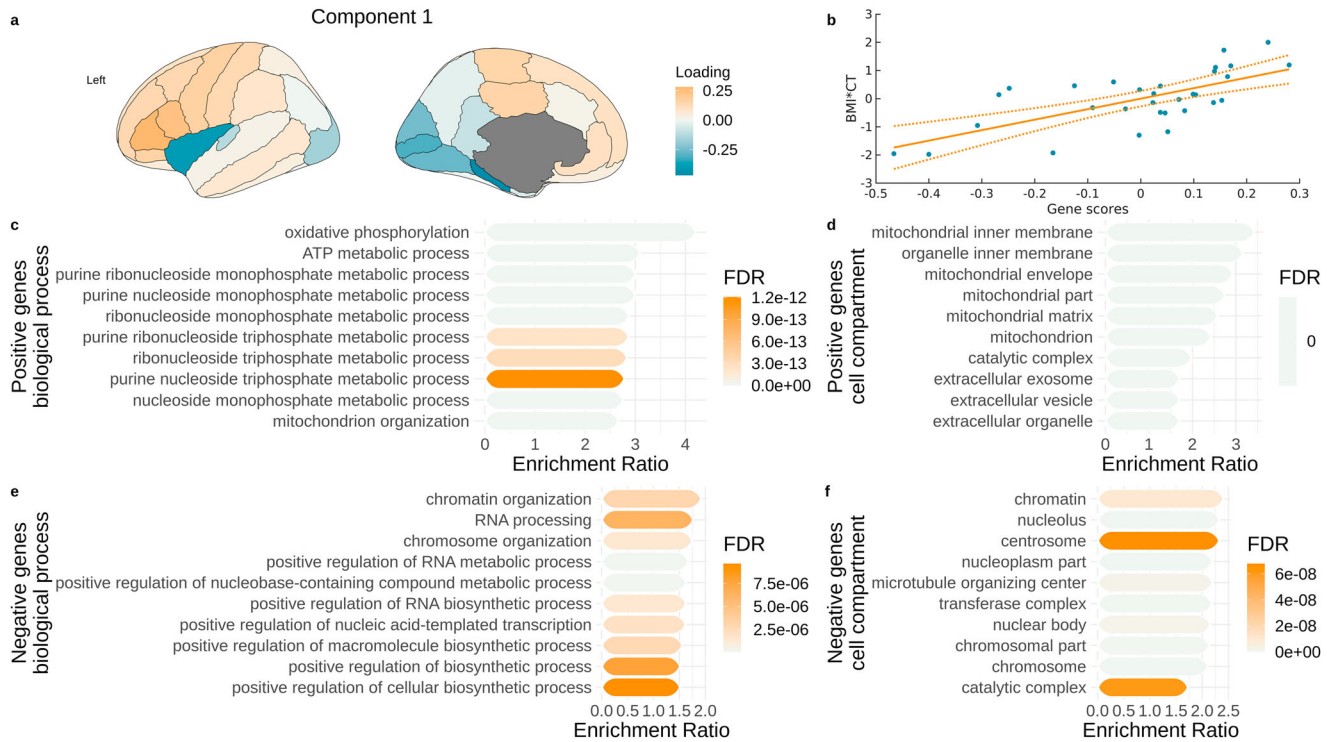

**Fig. 2 | Associations between gene expression and obesity map in the HCP-A sample. a** Gene scores plotted on the brain showing a summary map of gene expression related to BMI-CT relationship in the HCP-A sample; **b** Relationship between BMI-CT values and gene scores from the PLS analysis; **c–f** Gene overexpression analysis terms associated with sets of significant genes in the PLS analysis in the HCP-A sample. BMI - body mass index. CT cortical thickness, FDR false discovery rate.

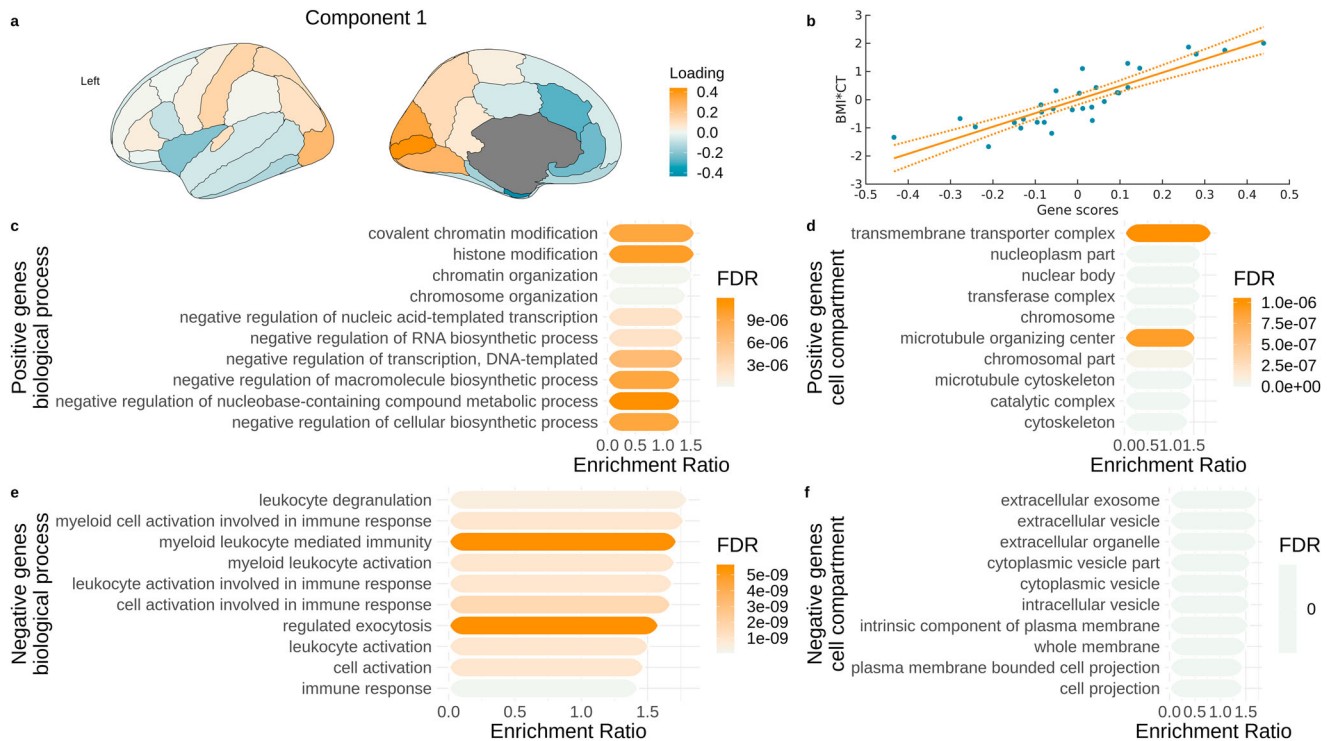

**Fig. 3 | Associations between gene expression and obesity map in the UKBB sample. a** Gene scores plotted on the brain showing a summary map of gene expression related to BMI-CT relationship in the UKBB sample; **b** Relationship between UKBB values and gene scores from the PLS analysis; **c–f** Gene overexpression analysis terms associated with sets of significant genes in the PLS analysis in the UKBB sample. BMI body mass index, CT cortical thickness, FDR false discovery rate.

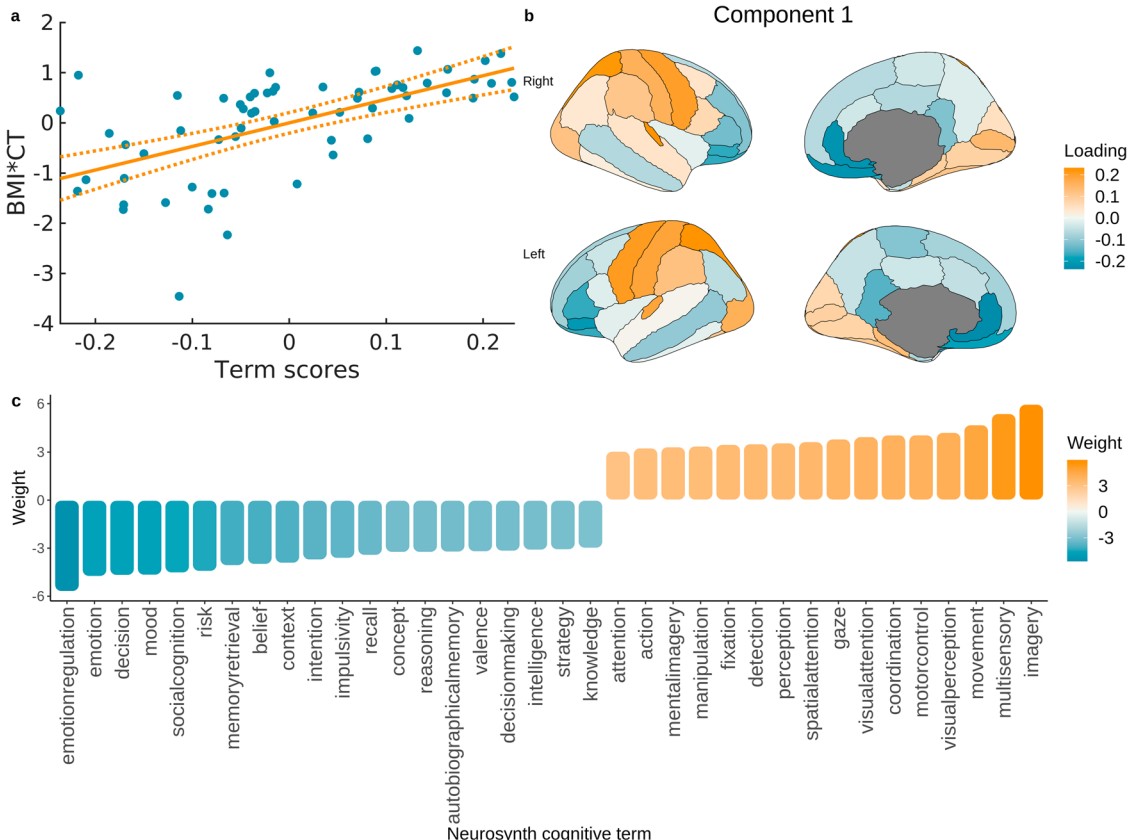

**Fig. 4 | Associations between cognitive brain maps and obesity map in the ABCD sample. a** Relationship between BMI-CT values and term scores from the PLS analysis; **b** term scores plotted on the brain showing a summary map of cognitive terms related to BMI-CT relationship in the ABCD sample; **c** cognitive terms significantly associated with obesity map in the ABCD sample (colours correspond to positive and negative associations with obesity maps as depicted in **b**). BMI body mass index, CT cortical thickness.

components that maximise covariance between 2 datasets - here genes from the AHBA/BrainSpan and obesity map per sample. Each latent component thus represents a gene expression pattern (gene weights) that is related to the obesity brain map. Significance of each latent component was calculated using permutation tests ($n = 10,000$) used to create a null distribution while accounting for spatial autocorrelation between cortical thickness data based on scripts at https://github.com/KirstieJane/NSPN_WhitakerVertes_PNAS2016/ and https://github.com/gecthomas/QSM_and_AHBA_transcription_in_PD[78–80]. In each PLS analysis, we tested 5 latent components. The strength of contribution of each gene to the latent components was assessed using bootstrapping (randomly resampling rows of brain and genetic data matrices with replacement; $n = 20,000$) and expressed as bootstrap ratios (BR) - a ratio of the original gene weights to the standard error estimated from bootstrapping. BRs are akin to z-scores under some conditions[81,82]. Genes with absolute values of BR higher than 3 (comparable to $p < 0.001$) were deemed as robustly contributing to the latent components and were retained for further analyses. Individual gene weights obtained from PLS for each latent component were summarised for each brain parcel and expressed as gene scores, thus yielding a brain map of a latent component-related gene expression pattern. If the general correlation between gene scores and obesity maps is positive, then the genes positively contributing to the latent component represent sets with higher expression in brain areas positively related to BMI, while the opposite is true for the genes negatively associated with the latent components.

**Gene overrepresentation analysis.** Here, we investigated biological processes and cellular components that were related to the gene sets associated with obesity maps for each sample. To this end, we performed

statistical overrepresentation analysis using the WEB-based GEne SeT AnaLysis Toolkit (http://webgestalt.org)[83]. All genes considered in our analyses were used as a reference list and lists of the most robustly contributing genes were used as data of interest. Lists were created separately for genes positively and negatively associated with latent components. False discovery rate (FDR) correction as implemented in WebGestalt was used to correct for multiple comparisons in the gene overrepresentation analysis. For ease of interpretation, we considered 10 processes/components most significantly associated with the gene sets.

**Cognitive process-cortical thickness spatial correspondence analysis.** As the last step, we tested how functional brain activations for 123 different cognitive processes correspond to obesity maps. Here, we used PLS analysis similarly to the previous step, where the gene matrix was substituted with a matrix of cognitive processes and their activation likelihood for each parcel of the DKT atlas. The remaining steps were identical to those in section 'Gene-cortical thickness spatial correspondence analysis'. Because the Neurosynth database does not differentiate between activations and deactivations, we cannot interpret the directionality of associations between obesity and cognitive maps.

## Results
### Associations between cortical thickness and BMI
We performed general linear model analyses to explore the correlations between cortical thickness and BMI across participants in 4 different age groups. Overall, we found negative associations with BMI in the ABCD sample, with the most pronounced effects in the fronto-temporal regions (Supplementary Data 1–4, Fig. 1). Throughout the ageing process, in all tested samples, we observed a negative relationship between BMI and

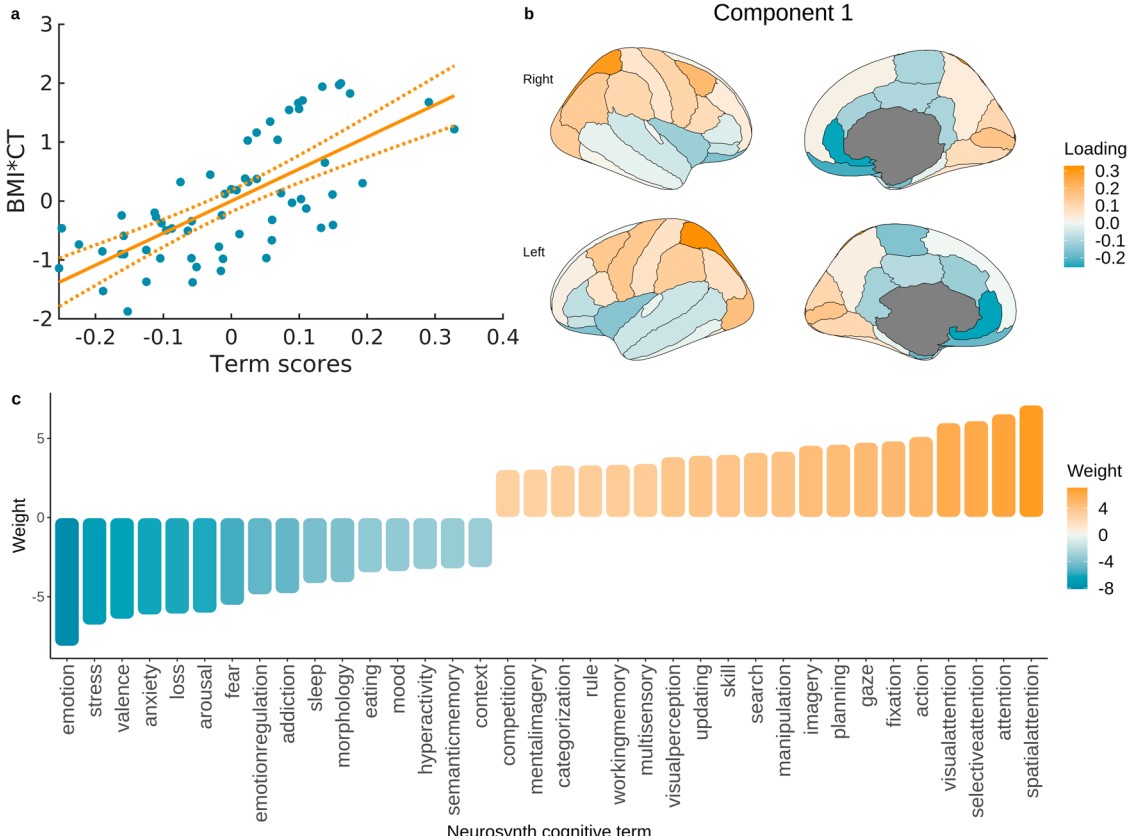

**Fig. 5 | Associations between cognitive brain maps and obesity map in the UKBB sample. a** Relationship between BMI-CT values and term scores from the PLS analysis; **b** term scores plotted on the brain showing a summary map of cognitive terms related to BMI-CT relationship in the UKBB sample; **c** cognitive terms significantly associated with obesity map in the UKBB sample (colours correspond to positive and negative associations with obesity maps as depicted in **b**). BMI body mass index, CT cortical thickness.

cortical thickness in the temporal and inferior frontal brain regions. Further, in adult samples, we observed some positive associations between BMI and cortical thickness, predominantly in the dorsal frontal and occipital regions of the brain. The effects of BMI on cortical thickness were almost identical when socioeconomic status was not regressed out (Fig. S2).

## Correspondence between obesity maps and neurotransmitter systems

We compared obesity maps in different age groups with neurotransmitter maps from ref. 22 and found no significant associations in the ABCD, HCP, or HCP-A sample. We found significant negative correlations between obesity maps and dopamine transporter, D2 receptor, vesicular acetylcholine transporter, and serotonin 1a receptor, and positive associations with GABAa receptor and serotonin 1b receptor in the UKBB sample (Fig. 1, Fig. S3, Supplementary Data 5). Supplementary analysis correlating age and association strength between PET data and obesity maps showed that age of the PET study samples did not influence the correlations with different age groups of the tested cohorts (Fig. S4).

## Correspondence between obesity maps and gene expression

Using PLS, we identified latent components relating obesity maps to gene expression maps from the AHBA and BrainSpan atlases. We did not find any significant components for the ABCD or HCP samples (with either AHBA or BrainSpan atlases). We found one significant component for the HCP-A sample ($p < 0.001$, $r = 0.654$; Fig. 2) that positively related the cortical thickness-BMI map to a latent component of gene expression patterns. This component explained 42.75% of variance in the brain data. Similarly, we found one significant component for the UKBB sample ($p < 0.001$, $r = 0.869$; Fig. 3). This component explained 75% of variance in the brain data. All remaining components did not reach statistical significance. Positive correlations between gene scores and obesity brain maps mean that genes negatively related to the component (negative genes) were expressed to a higher degree in brain regions with negative BMI-cortical thickness associations and to a lower degree in brain regions with positive BMI-cortical thickness associations, with an opposite relationship for genes positively related to the component (positive genes). Overall, we found increasing correspondence between obesity maps and gene expression with increasing age of the tested sample.

## Gene overexpression analysis

Using the WebGestalt tool[83], we analysed the sets of genes associated with significant PLS components from the previous step to interpret biological processes and cellular components to which the gene expression is related. In the HCP-A sample, lower cortical thickness (negative genes) was significantly associated with regulation of biosynthetic processes, chromosome organisation, and chromosomal cell components, while higher cortical thickness (positive genes) was associated with metabolic processes, creation of ATP, and mitochondria (Fig. 2). In the UKBB sample, lower cortical thickness (negative genes) was associated with inflammation-related processes, such as leukocyte degranulation or general immune response, while higher cortical thickness (positive genes) was related to biosynthetic and metabolic processes, chromatin organisation, and cytoskeleton, chromosome, and nuclear body cell components (Fig. 3).

## Correspondence between obesity maps and brain signatures of cognitive processes

Significant associations between obesity maps and Neurosynth cognitive maps were found in the ABCD and UKBB samples. In the ABCD sample,

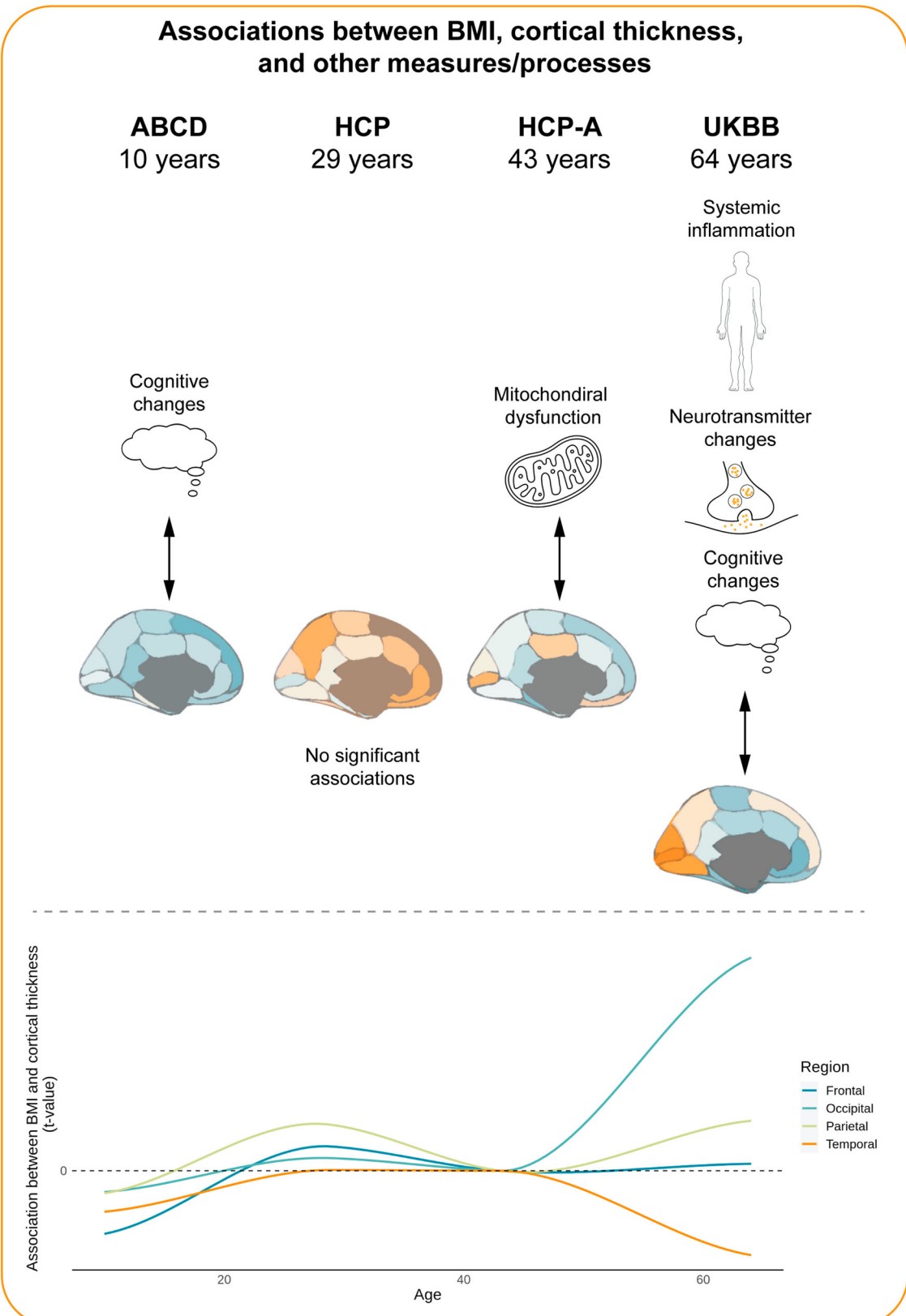

**Fig. 6 | Summary figure reflecting obesity-related brain changes in all samples.** Average *t*-values per lobe are depicted in the line plot.

the first component explained 36% of variance in the brain data ($p < 0.001$, $\rho = 0.601$; Fig. 4). Cognitive processes related to obesity-associated brain changes in this sample are depicted in Fig. 4 and include visual perception, attention, intelligence, decision making, impulsivity, risk taking, and

emotion regulation. In the UKBB sample, the first component explained 48.5% variance in the brain data ($p < 0.001$, $\rho = 0.697$). Cognitive processes identified in this analysis are depicted in Fig. 5 and include processes similar to the ones identified in the ABCD sample - spatial attention, visual

perception, working memory, emotion regulation, fear, arousal, and anxiety. In the HCP and HCP-A samples we did not find any significant PLS components for cognitive processes ($p > 0.05$).

## Discussion

Neuroanatomical and cognitive correlates of obesity change across the lifespan. In this study, we aimed to identify structural (cortical thickness), genetic, neurochemical, and cognitive underpinnings of obesity in different age groups. We showed that obesity is consistently related to lower cortical thickness in the temporal and inferior frontal brain regions across all age groups, with the associations becoming more pronounced with age. We also found positive relationships between cortical thickness and BMI in adults, predominantly in the dorsal frontal and occipital brain regions. In middle-aged adults, positive associations were related to gene expression related to mitochondrial processes, while in older adults, lower cortical thickness was related to inflammatory processes. We also showed here that cortical thickness changes in human obesity share some similarities with brain distributions of serotonin, dopamine, acetylcholine, and GABA neurotransmitter systems, and that these associations are only visible in older adults. Finally, we found that in children and older adults, brain changes related to obesity are located in areas involved in emotional and attentional cognitive processes (Fig. 6). Overall, and in keeping with previous studies, our results show different and more pronounced brain changes related to obesity with increasing age[6,10–12,16,84].

We investigated lifespan changes of brain-obesity associations by testing samples of different ages. Our premise is that certain brain features in obesity underpin behaviours that lead to over-eating and excess weight gain, and that these could be present throughout life. On the other hand, negative effects of chronic adiposity on the brain should accumulate over the lifespan[5,6,12]. Our findings of associations between cortical thickness and obesity are mostly in line with these hypotheses, given that we found predominantly fronto-temporal cortical thinning, increasing with age. These findings are also supported by studies investigating grey matter volume and surface area showing that across different age groups obesity is negatively related to changes predominantly in the frontal but also temporal cortices[6–8,12,84–89].

While we cannot rule out adiposity-related neurodegeneration in the 10–11-year-old ABCD cohort, cortical thickness effects in children could also reflect differences in brain maturation, which may underpin a vulnerability factor for obesity through behavioural changes, as shown previously[6]. Conversely, higher cortical thinning with age in adults might point to an ongoing process of neurodegeneration associated with excess weight and related comorbidities[14]. According to our gene expression analyses, these changes in the brain in adults might be specifically related to mitochondrial function and inflammatory processes. This is in line with a large body of literature suggesting that obesity is related to systemic inflammation, which leads to neurodegeneration and could be related to mitochondrial dysfunction[12,90,91].

Neurotransmitter systems related to obesity in adulthood in this study have been previously reported as involved in control of food intake and obesity in humans and animals[92–99]. While spatial associations do not prove a mechanistic relationship between neurotransmitter signalling and obesity, our findings suggest some degree of involvement of those neurotransmitters - as a cause or a consequence. One reason for associations of those systems with obesity being present only in the oldest age group might be that they are a consequence of an obesity-related neurodegeneration that increases with age. Ageing is associated with neural changes within distinct neurotransmitter systems, and this might be accentuated by obesity[100,101]. Speculatively, this finding could suggest a vicious cycle, where obesity-related brain pathology leads to changes in the brain that promote further increased food intake and thus further brain pathology.

Finally, we characterised cognitive correlates of obesity in children and older adults, showing that obesity-related cortical thickness changes are associated with cognitive processes predominantly involved in attention and emotional processing. Previously, obesity and risk for obesity in childhood were associated with altered attention for food and non-food stimuli and emotional processing and eating compared to control groups[102–104]. In adults, emotional processing is often associated with obesity in the form of a phenotype labelled uncontrolled eating[105]. At the same time, adult individuals with obesity show attentional bias for food that seems to predict weight gain[106,107]. Together, attentional and emotional processes alongside their underlying neural correlates might constitute a vulnerability factor for weight gain that remains stable across lifespan.

Some limitations of this study have to be considered: while the ABCD and UKBB samples were large, the HCP and HCP-A samples were smaller and the results might therefore be less reliable. In addition, main analyses compared obesity maps with gene expression, neurotransmitter, and cognitive maps from other samples that differed in age, sex distribution, and likely other factors that we did not account for. These analyses are purely correlational and need to be interpreted with caution. Importantly, all analyses were performed using the DKT atlas which includes 62 cortical parcels and might therefore not distinguish between certain smaller specialised cortical areas that might be pertinent in obesity research. Relatedly, we did not investigate the involvement of subcortical structures in obesity across different ages, which, given the subcortical contributions to obesity[12,86,88], is an important topic for future investigations.

In sum, we show that brain and genetic correlates of obesity change across the lifespan, supporting our hypothesis that BMI-related neural endophenotypes might have a different significance in children and adults. This is a cross-sectional study, so causality cannot be inferred. However, in connection with previous studies we believe that accelerated brain maturation in children might increase the risk for obesity - potentially through altered emotional and attentional processing - while in adults, obesity might lead to brain degeneration that involves mitochondrial function and neuroinflammation. This study contributes to the knowledge necessary to understand obesity and create targeted prevention and intervention strategies.

## Data availability

Data used in this study are available through the ABCD, HCP, and UKBB consortia upon request.

## Code availability

Code used to analyse the datasets was deposited at https://doi.org/10.5281/zenodo.11068535[72]. and at https://github.com/FilipMorys/Obesity_maps.

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

## Acknowledgements

This work was supported by a Foundation Scheme award to AD from the Canadian Institutes of Health Research, Fonds de recherche du Québec-Santé postdoctoral fellowship to FM. A portion of the data used in the preparation of this article were obtained from the Adolescent Brain Cognitive Development SM (ABCD) Study (https://abcdstudy.org), held in the NIMH Data Archive (NDA). This is a multisite, longitudinal study designed to recruit more than 10,000 children aged 9–10 and follow them over 10 years into early adulthood. The ABCD Study® is supported by the National Institutes of Health and additional federal partners under award numbers U01DA041048, U01DA050989, U01DA051016, U01DA041022, U01DA051018, U01DA051037, U01DA050987, U01DA041174, U01DA041106, U01DA041117, U01DA041028, U01DA041134, U01DA050988, U01DA051039, U01DA041156, U01DA041025, U01DA041120, U01DA051038, U01DA041148, U01DA041093, U01DA041089, U24DA041123, U24DA041147. A full list of supporters is available at https://abcdstudy.org/federal-partners.html. A listing of participating sites and a complete listing of the study investigators can be found at https://abcdstudy.org/consortium_members/. ABCD consortium investigators designed and implemented the study and/or provided data but did not participate in the analysis or writing of this report. This manuscript reflects the views of the authors and may not reflect the opinions or views of the NIH or ABCD consortium investigators. The ABCD data repository grows and changes over time. The ABCD Study data used in this report came from https://doi.org/10.15154/zw9a-7t30. Data were provided in part by the Human Connectome Project, WU-Minn Consortium (Principal Investigators: David Van Essen and Kamil Ugurbil; 1U54MH091657) funded by the 16 NIH Institutes and Centers that support the NIH Blueprint for Neuroscience Research; and by the McDonnell Center for Systems Neuroscience at Washington University. This research has been conducted using data from UK Biobank, a major biomedical database.

## Author contributions

F.M. and A. Dagher designed the study; F.M., C.T., S.R., J.Y.H., and A. Dai analysed the data; all authors took part in results interpretation and manuscript preparation.

## Competing interests

The authors declare no competing interests.
