## [Peer Review File · Communications Biology]

Reviewers' comments:

Reviewer #1 (Remarks to the Author):

Morys and colleagues investigate the association between cortical thickness and body mass index across the lifespan in large-scale datasets. They identify lower cortical thickness in fronto-temporal brain regions with obesity, which coincides with previous findings. Moreover, Morys et al. investigated the relationship between cortical thickness and obesity with respect to cognitive processes, gene expression, and neurotransmitter systems.

The study addresses a very timely question and makes use of large-scale data sets with more than 45 000 participants.

The authors make the point that investigating multiple cohorts can help to understand causality (i.e. Effect of obesity on brain structure). How so? Each of the data sets, on different age groups, were investigated separately and no longitudinal data was used (even though it should be available from the ABCD sample). In the current form, I find this an overinterpretation of the findings. Analyzing the cortical thickness datasets of the different cohorts together (with the same fsl version and same quality criteria) would greatly improve the manuscript and increase novelty. Is this possible?

The data was processed partly with different freesurfer versions. The data for the adolescents were processed with freesurfer 5.3.0, while the other data bases were processed with freesurfer 6.0.0 or 6.0.1.

How was this taken into account?

The criteria for quality control might not be consistent for the different data bases. To what extent is the quality control comparable between the different data sets. What criteria were used?

Why not reanalyze all data in the same pipeline using the same fsl version.

Please provide more information in the method section with respect to the MRI measurements.

Mention the type of sequence used (i.e. T1 etc.).

Was is meant by "neuromaps" in section 2.3

The authors report that they used Neurosynth to map the voxels associated with cognitive functions.

How many studies were involved in the meta-analyses? Please add the corresponding map.

Is it necessary to correlate with 123 cognitive processes? There should be a strong redundancy and correlation between many of these processes.

Since age has a significant impact on brain structure, it would of interest to evaluate the interaction between BMI and age. Based on the summary figure (fig 6), I would hypothesize that there is a significant interaction between BMI and age, especially in adults.

Figure 1: Abbreviations of Neurotransmitter systems need to be explained in the figure legend. Add age range next to cohort name, next to a, b, c, d

Discussion: in the current form, I do not see how the authors can link the identified associations to causality.

Reviewer #2 (Remarks to the Author):

In their manuscript entitled "Neural correlates of obesity across the lifespan", Morys and colleagues investigated the morphometric, neurochemical, genetic, and cognitive correlates of obesity across different age groups.

To this aim, the authors took advantage of large public repositories of MRI data to study the associations between BMI and cortical thickness in children, young adults, middle aged adults, and older adults. To further qualify the neurostructural changes associated with BMI across the lifespan, neuroimaging data were then correlated with data on gene expression (AHBA), neurotransmitter receptor and transporter distribution (PET data), and cognitive processes (Neurosynth). Across the age groups, BMI was negatively associated with cortical thickness in inferior frontal and temporal regions, and these associations were more pronounced with increasing age; in adults, BMI was positively associated with cortical thickness in dorsal frontal and occipital regions. In middle-aged adults, positive changes were associated with gene expression related to mitochondrial processes, whereas in older adults negative associations were linked to inflammatory processes. Further, cortical thickness changes associated with obesity overlapped with the brain distribution of several neurotransmitter systems, including serotonin, dopamine, acetylcholine, and GABA. Finally, in children and older adults, brain changes occurred in areas involved in emotional and attentional cognitive processes.

The authors conclude that brain changes associated with obesity in children are more likely to reflect vulnerability factors, whereas changes in older adults may reflect neurodegeneration associated with the obesity status and related comorbidities.

The paper is well written, the study is methodologically sound, and the results represent a relevant addition to the field. In particular, the authors provide a first attempt to bridge the gap between different levels of explanation of obesity, including genetic, neurochemical, brain morphometric, and cognitive. Despite correlational, the results point to a model whereby early (neuro)cognitive changes favor overeating, leading to obesity-related brain pathology later in life; this, in turn, promotes food intake and worsens brain/system pathology.

Minor revisions

While I have no major concerns, the authors may want to consider the following recommendations:

1. Discussion of neuroanatomical correlates of obesity. Integrating previous findings on the neuroanatomical correlates of obesity investigated with a different method may be beneficial to this section. For example, recent meta-analyses of voxel-based morphometry studies performed by several research groups (e.g., García-García et al., 2019; Herrmann et al., 2019; Chen et al., 2020; Zapparoli et al., 2022) showed that obesity is consistently associated with reduced gray matter volume in the prefrontal cortex, including the orbitofrontal cortex. If anything, this line of evidence suggests that different methods and measures of the neuromorphometric changes associated with obesity converge in the prefrontal cortex.
2. Figure 6. The figure's caption should be expanded to provide more information for the correct interpretation of the figure. For example, what does represent the dotted horizontal line in the bottom part of the figure? What information is plotted on the y-axis (e.g., arbitrary units, average estimate values within cortical lobes)?
3. Table S1-S4. The information on the laterality (left vs. right hemisphere) of the cortical parcel should be added to the tables. To improve the readability, the parcels showing a significant association with BMI should be highlighted.
4. Limitations. By design, the authors focused their analyses on (62) cortical parcels of the DKT atlas. This choice is justified by several reasons, including consistency with other data samples and analyses. While valid, this approach may miss out the functional specialization that occurs within extended and functionally heterogenous cortical areas, such as the insular cortex. In the same vein, the authors may want to acknowledge the contribution of subcortical areas (e.g., amygdala, thalamus, mesencephalic nuclei; Herrmann et al. 2019, Zapparoli et al. 2022) to overeating and obesity, something not addressed here but that could be investigated in future studies.

References

Chen, E. Y., Eickhoff, S. B., Giovannetti, T., & Smith, D. V. (2020). Obesity is associated with reduced orbitofrontal cortex volume: A coordinate-based meta-analysis. *NeuroImage: Clinical*, 28, 102420.

García-García, I., Michaud, A., Dadar, M., Zeighami, Y., Neseliler, S., Collins, D. L., ... & Dagher, A. (2019). Neuroanatomical differences in obesity: meta-analytic findings and their validation in an independent dataset. *International Journal of Obesity*, 43(5), 943-951.

Herrmann, M. J., Tesar, A. K., Beier, J., Berg, M., & Warrings, B. (2019). Grey matter alterations in obesity: A meta-analysis of whole-brain studies. *Obesity reviews*, 20(3), 464-471.

Zapparoli, L., Devoto, F., Giannini, G., Zonca, S., Gallo, F., & Paulesu, E. (2022). Neural structural abnormalities behind altered brain activation in obesity: Evidence from meta-analyses of brain activation and morphometric data. *NeuroImage: Clinical*, 36, 103179.

Reviewer #3 (Remarks to the Author):

This descriptive study uses available large neuroimaging datasets and gene expression, neurotransmitter, and cognitive data at 4 different age cohorts to evaluate correlations between obesity and the brain across the lifespan. The multi-faceted approach is innovative, and the manuscript provides important information about the impact of obesity at different ages across the lifespan. Findings from the gene overexpression analysis that obesity correlates with differential gene expression related to inflammation and mitochondrial function in adults and older adults are novel and corroborate prior work in the field. The findings showing that brain changes related to obesity are in areas involved in emotional and attention in children and older adults are of interest and warrant further investigation. The correlative nature of this work prohibits the ability to understand cause and effect. Nonetheless, the data are hypothesis-generating and could lead to future studies to uncover underlying mechanisms. The findings would be of interest to researchers in the field of obesity and the brain.

Specific Comments:

The smaller sample sizes of the HCP compared to ABCD or UK Biobank make interpreting age-related differences in obesity-neural relationships difficult.

The figures are difficult to interpret. Orientation of the brain images is necessary. In addition, identifying the brain areas where negative correlations or positive correlations between BMI and cortical thickness (in Figure 1) were identified in each cohort would be helpful.

The supplemental tables (S1-S4) show patterns of negative associations between BMI and cortical thickness and no significant positive associations in the ABCD cohort. In contrast, the HCP cohort shows primarily positive associations between BMI and cortical thickness, HCP-A has somewhat weak associations, likely due to small sample sizes, and the UK Biobank shows strong negative and positive associations. The interpretation of these findings was summarized as "lower cortical thickness in frontotemporal brain regions associated with obesity across all age cohorts," which seems overly simplified.

Figure 6 visually depicts the relationship between BMI and cortical thickness across all age groups. The figure shows a positive relationship between BMI and cortical thickness in the younger age group, but the data show otherwise. Please clarify.

We would like to thank the editor and reviewers for the evaluation of our manuscript and their comments. Below we present a point-by-point response. All answers have been marked in italics and new text added to the manuscript has been marked in red.

Reviewer #1 (Remarks to the Author):

Morys and colleagues investigate the association between cortical thickness and body mass index across the lifespan in large-scale datasets. They identify lower cortical thickness in fronto-temporal brain regions with obesity, which coincides with previous findings. Moreover, Morys et al. investigated the relationship between cortical thickness and obesity with respect to cognitive processes, gene expression, and neurotransmitter systems.

The study addresses a very timely question and makes use of large-scale data sets with more than 45 000 participants.

The authors make the point that investigating multiple cohorts can help to understand causality (i.e. Effect of obesity on brain structure). How so? Each of the data sets, on different age groups, were investigated separately and no longitudinal data was used (even though it should be available from the ABCD sample). In the current form, I find this an overinterpretation of the findings. Analyzing the cortical thickness datasets of the different cohorts together (with the same fsl version and same quality criteria) would greatly improve the manuscript and increase novelty. Is this possible?

We would like to thank the reviewer for this comment and the overall evaluation of our manuscript. We mentioned that our investigation of samples with different ages might help to establish causality, however, after reviewers' comments, we removed the parts directly inferring causal associations. This is because, as this reviewer points out, the data are cross-sectional. Nonetheless, we believe our results showing different brain-obesity associations in different age groups are novel.

We agree that analysing cortical thickness of the datasets together is of interest, however we do not think it helps our aims here. First, the main source of difference between datasets is not the software version but the MRI acquisition parameters and scanners. Using the same freesurfer version would not remove these effects. Also, here we do not compare brain anatomy across studies - something that may not be possible in any case. Rather we use each dataset as a self-contained study, eliminating the issue of acquisition or software version differences. We note that we did not attempt to quantitatively investigate age-related differences. Rather, our study aims to qualitatively compare the 4 different samples.

Finally, investigating the samples used in this study together might not be feasible due to age gaps between datasets, e.g. ABCD and HCP.

The data was processed partly with different freesurfer versions. The data for the adolescents were processed with freesurfer 5.3.0, while the other data bases were processed with freesurfer 6.0.0 or 6.0.1.

How was this taken into account?

This is a very important point that we considered prior to conducting the study. Indeed different software versions might significantly influence outcome measures. However, according to Haddad and colleagues, FreeSurfer versions 5.3 and 6.0 produce similar output in terms of cortical thickness measures [1]. In addition, given that the comparison between samples is qualitative, not quantitative, we believe that the usage of different versions of FreeSurfer does not significantly affect our conclusions. As mentioned earlier, scanner and acquisition differences are a greater source of variability than software version.

The criteria for quality control might not be consistent for the different data bases. To what extent is the quality control comparable between the different data sets. What criteria were used?
Why not reanalyze all data in the same pipeline using the same fsl version.

Quality control (QC) procedures indeed differ slightly between cohorts. For the ABCD, QC involved pre- and post-processing visual inspections performed by the ABCD consortium [2]. HCP and HCP aging samples are similar in this respect, in that they are inspected pre processing by the HCP consortia and the FreeSurfer output was visually inspected by us. In terms of the UKBB dataset, automatic QC is implemented that uses algorithms to identify images that are not of satisfying quality - both pre- and post-processing [3]. Nonetheless, strict QC that was implemented in all of the investigated samples should correctly exclude individuals with pre- and post-processing issues.

Please see our response to comment #1 as to re-analysis of the data altogether with the same version of FreeSurfer.

Please provide more information in the method section with respect to the MRI measurements. Mention the type of sequence used (i.e. T1 etc.).

We thank the reviews for this suggestion. We have now added a brief description to each neuroimaging section of the manuscript informing that we indeed used a T1-weighted sequence and stating the voxel size. We believe that this is the most important information and the remaining details of MRI sequences can be easily found in the cited literature.

Some of the added text now reads:

“Structural T1-weighted images with 1mm³ isotropic voxel size were used to obtain cortical morphometry measures.”

This information is added to the description of all four imaging datasets.

Was is meant by “neuromaps” in section 2.3

Neuromaps is a python software package developed by Markello and colleagues that includes curated neuro-receptor data and enables researchers to investigate how they relate to other brain phenotypes. We now included a brief description in the manuscript text that reads:

“ ‘Neuromaps’ is a toolbox that contains several curated brain maps and software tools that allows researchers to make comparisons between different brain maps.”

The authors report that they used Neurosynth to map the voxels associated with cognitive functions. How many studies were involved in the meta-analyses? Please add the corresponding map.

Neurosynth utilises a set of over 15,000 research papers to identify brain correlates of cognitive processes, amongst others. The final sample size for each cognitive process identified in this study is therefore different and can be found directly on neurosynth. We have now added brief information on data availability - cognitive processes maps previously obtained by Hansen and colleagues [4] are available at a github link specified below.

“Maps of all cognitive processes can be found at https://github.com/netneurolab/hansen_receptors. List of included cognitive processes can be found in the supplement to [4].”

Is it necessary to correlate with 123 cognitive processes? There should be a strong redundancy and correlation between many of these processes.

The 123 processes used in this study were chosen from an atlas of cognitive processes and follow already published literature [4]. It is likely that there are correlations between some cognitive processes, and hence their brain correlates. Partial least square analysis accounts for correlation amongst the input measures. This is done at no cost to statistical power, given the nature of partial least square analysis that finds linear combinations of certain variables to maximise correlations between two data matrices.

Since age has a significant impact on brain structure, it would of interest to evaluate the interaction between BMI and age. Based on the summary figure (fig 6), I would hypothesize that there is a significant interaction between BMI and age, especially in adults.

We agree with this reviewer that this is an important topic. In fact, the premise of our paper is exactly this - existing interactions between BMI and age that differently influence the brain. In this study, we decided to conduct a qualitative comparison of such interactions. We agree that a quantitative assessment is of interest and we respond in our response to comment #1 why we did not conduct such analyses in this particular study. We also note that for example the age range in the ABCD sample might not be sufficient enough to conduct such an investigation within single samples. Ideally, more samples should be acquired and used in a quantitative assessment of age and BMI interaction.

Figure 1: Abbreviations of Neurotransmitter systems need to be explained in the figure legend. Add age range next to cohort name, next to a, b, c, d

We thank the reviewer for this suggestion. The changes have been implemented:

*“Figure 1 - Obesity maps and their relationship with neurotransmitter systems. a Relationship between BMI and cortical thickness in the ABCD sample; b Relationship between BMI and cortical thickness in the HCP sample; c Relationship between BMI and cortical thickness in the HCP-A sample; d Relationship between BMI and cortical thickness in the UKBB sample; e correlations between obesity maps and neurotransmitter systems (chosen neurotransmitter maps derived from largest samples - see Figure S2 for all maps); * denotes significant associations. BMI - body mass index. CT - cortical thickness. GABA_A - gamma-aminobutyric acid receptor a. 5HT_{1b} - serotonin 1b receptor. M₁ - muscarinic M1 receptor. NET - norepinephrine transporter. 5HT₆ - serotonin 6 receptor. α₄β₂ NR - α₄β₂ nicotinic receptor. mGluR₅ - metabotropic glutamate receptor 5. 5HT_{2a} - serotonin 2a receptor. D₁ - dopamine D1 receptor. H₃ - histamine H3 receptor. 5HTT - serotonin transporter. VACHT - vesicular acetylcholine transporter. CB₁ - cannabinoid receptor 1. 5HT₄ - serotonin 4 receptor. DAT - dopamine transporter. MOR - μ-opioid receptor. D₂ - dopamine D2 receptor. 5HT_{1a} - serotonin 1a receptor.”*

Discussion: in the current form, I do not see how the authors can link the identified associations to causality.

We thank the reviewer for this comment. As mentioned earlier, we removed all suggestions about causality from the manuscript.

Reviewer #2 (Remarks to the Author):

In their manuscript entitled “Neural correlates of obesity across the lifespan”, Morys and colleagues investigated the morphometric, neurochemical, genetic, and cognitive correlates of obesity across different age groups.

To this aim, the authors took advantage of large public repositories of MRI data to study the associations between BMI and cortical thickness in children, young adults, middle aged adults, and older adults. To further qualify the neurostructural changes associated with BMI across the lifespan, neuroimaging data were then correlated with data on gene expression (AHBA), neurotransmitter receptor and transporter distribution (PET data), and cognitive processes (Neurosynth).

Across the age groups, BMI was negatively associated with cortical thickness in inferior frontal and temporal regions, and these associations were more pronounced with increasing age; in adults, BMI was positively associated with cortical thickness in dorsal frontal and occipital regions. In middle-aged adults, positive changes were associated with gene expression related to mitochondrial processes, whereas in older adults negative associations were linked to inflammatory processes. Further, cortical thickness changes associated with obesity overlapped with the brain distribution of several neurotransmitter systems, including serotonin, dopamine, acetylcholine, and GABA. Finally, in children and older adults, brain changes occurred in areas involved in emotional and attentional cognitive processes.

The authors conclude that brain changes associated with obesity in children are more likely to reflect vulnerability factors, whereas changes in older adults may reflect neurodegeneration associated with the obesity status and related comorbidities.

The paper is well written, the study is methodologically sound, and the results represent a relevant addition to the field. In particular, the authors provide a first attempt to bridge the gap between different levels of explanation of obesity, including genetic, neurochemical, brain morphometric, and cognitive. Despite correlational, the results point to a model whereby early (neuro)cognitive changes favor overeating, leading to obesity-related brain pathology later in life; this, in turn, promotes food intake and worsens brain/system pathology.

Minor revisions

While I have no major concerns, the authors may want to consider the following recommendations:

1. Discussion of neuroanatomical correlates of obesity. Integrating previous findings on the neuroanatomical correlates of obesity investigated with a different method may be beneficial to this section. For example, recent meta-analyses of voxel-based morphometry studies performed by several research groups (e.g., García-García et al., 2019; Herrmann et al., 2019; Chen et al., 2020; Zapparoli et al., 2022) showed that obesity is consistently associated with reduced gray matter volume in the prefrontal cortex, including the orbitofrontal cortex. If anything, this line of evidence suggests that different methods and measures of the neuromorphometric changes associated with obesity converge in the prefrontal cortex.

We would like to thank the reviewer for the overall evaluation of our manuscript. We have now expanded our discussion section to include suggested information and strengthen our argumentation. The added section reads:

“These findings are also supported by studies investigating grey matter volume and surface area showing that across different age groups obesity is related to changes predominantly in the frontal but also temporal cortices [5–14].”

2. Figure 6. The figure's caption should be expanded to provide more information for the correct interpretation of the figure. For example, what does represent the dotted horizontal line in the bottom part of the figure? What information is plotted on the y-axis (e.g., arbitrary units, average estimate values within cortical lobes)?

We apologise for the omissions. We have added the information to the plot regarding what the dotted line represents (a t-value of 0) and what information is plotted on the y-axis. We also expanded the figure caption.

3. Table S1-S4. The information on the laterality (left vs. right hemisphere) of the cortical parcel should be added to the tables. To improve the readability, the parcels showing a significant association with BMI should be highlighted.

We apologise for not including the laterality information. This has now been changed and all significant associations are marked in bold. The same has been done in Table S5.

4. Limitations. By design, the authors focused their analyses on (62) cortical parcels of the DKT atlas. This choice is justified by several reasons, including consistency with other data samples and analyses. While valid, this approach may miss out the functional specialization that occurs within extended and functionally heterogenous cortical areas, such as the insular cortex. In the same vein, the authors may want to acknowledge the contribution of subcortical areas (e.g., amygdala, thalamus, mesencephalic nuclei; Herrmann et al. 2019, Zapparoli et al. 2022) to overeating and obesity, something not addressed here but that could be investigated in future studies.

We thank the reviewer for this important suggestion. We have implemented the suggested additions in the limitations paragraph:

“Importantly, all analyses were performed using the DKT atlas which includes 62 cortical parcels and might therefore not distinguish between certain smaller specialised cortical areas that might be pertinent in obesity research. Relatedly, we did not investigate the involvement of subcortical structures in obesity across different ages, which, given the subcortical contributions to obesity [6, 8, 9], is an important topic for future investigations.”

References

Chen, E. Y., Eickhoff, S. B., Giovannetti, T., & Smith, D. V. (2020). Obesity is associated with reduced orbitofrontal cortex volume: A coordinate-based meta-analysis. *NeuroImage: Clinical*, 28, 102420.

García-García, I., Michaud, A., Dadar, M., Zeighami, Y., Neseliler, S., Collins, D. L., ... & Dagher, A. (2019). Neuroanatomical differences in obesity: meta-analytic findings and their validation in an independent dataset. *International Journal of Obesity*, 43(5), 943-951.

Herrmann, M. J., Tesar, A. K., Beier, J., Berg, M., & Warrings, B. (2019). Grey matter alterations in obesity: A meta-analysis of whole-brain studies. *Obesity reviews*, 20(3), 464-471.

Zapparoli, L., Devoto, F., Giannini, G., Zonca, S., Gallo, F., & Paulesu, E. (2022). Neural structural abnormalities behind altered brain activation in obesity: Evidence from meta-analyses of brain activation and morphometric data. *NeuroImage: Clinical*, 36, 103179.

Reviewer #3 (Remarks to the Author):

This descriptive study uses available large neuroimaging datasets and gene expression, neurotransmitter, and cognitive data at 4 different age cohorts to evaluate correlations between obesity and the brain across the lifespan. The multi-faceted approach is innovative, and the manuscript provides important information about the impact of obesity at different ages across the lifespan. Findings from the gene overexpression analysis that obesity correlates with differential gene expression related to inflammation and mitochondrial function in adults and older adults are novel and corroborate prior work in the field. The findings showing that brain changes related to obesity are in areas involved in emotional and attention in children and older adults are of interest and warrant further investigation. The correlative nature of this work prohibits the ability to understand cause and effect. Nonetheless, the data are hypothesis-generating and could lead to future studies to uncover underlying mechanisms. The findings would be of interest to researchers in the field of obesity and the brain.

Specific Comments:

The smaller sample sizes of the HCP compared to ABCD or UK Biobank make interpreting age-related differences in obesity-neural relationships difficult.

We agree with the reviewer and have mentioned this as an important limitation of our study.

The figures are difficult to interpret. Orientation of the brain images is necessary. In addition, identifying the brain areas where negative correlations or positive correlations between BMI and cortical thickness (in Figure 1) were identified in each cohort would be helpful.

We thank the reviewer for this comment. We have added the orientation of the brain images to each figure in the manuscript. In Figure 1 we aimed to present full 'brain maps' of the associations between cortical thickness and BMI. This was done to represent the overall pattern of changes to facilitate the comparison between different cohorts. Tables S1-S4 include the actual significant associations that have been now bolded and hence the significance information is more accessible.

The supplemental tables (S1-S4) show patterns of negative associations between BMI and cortical thickness and no significant positive associations in the ABCD cohort. In contrast, the HCP cohort shows primarily positive associations between BMI and cortical thickness, HCP-A has somewhat weak associations, likely due to small sample sizes, and the UK Biobank shows strong negative and positive associations. The interpretation of these findings was summarized as "lower cortical thickness in frontotemporal brain regions associated with obesity across all age cohorts," which seems overly simplified.

We agree with the reviewer that this information in the abstract of our study might have been an oversimplification. We have now changed it and talk about different cortical thickness patterns in all samples:

"Our findings reveal a consistent pattern of lower cortical thickness in fronto-temporal brain regions associated with obesity across all age cohorts and varying age-dependent patterns in remaining brain regions."

In addition, in the manuscript we attempted to discuss the findings in their entirety. We note that we do mention the differences between samples in the discussion section.

"We showed that obesity is consistently related to lower cortical thickness in the temporal and inferior frontal brain regions across all age groups, with the associations becoming more

pronounced with age. We also found positive relationships between cortical thickness and BMI in adults, predominantly in the dorsal frontal and occipital brain regions.”

Figure 6 visually depicts the relationship between BMI and cortical thickness across all age groups. The figure shows a positive relationship between BMI and cortical thickness in the younger age group, but the data show otherwise. Please clarify.

We apologise for the lack of clarity in Figure 6. The dashed line in the lower panel depicts a value of 0, meaning that the average t-values found in children in all lobes are indeed negative, as the data suggests. We have now improved the figure.

References

1. Haddad E, Pizzagalli F, Zhu AH, Bhatt RR, Islam T, Ba Gari I, et al. Multisite test-retest reliability and compatibility of brain metrics derived from FreeSurfer versions 7.1, 6.0, and 5.3. *Hum Brain Mapp.* 2023;44:1515–1532.
2. Hagler DJ, Hatton SN, Cornejo MD, Makowski C, Fair DA, Dick AS, et al. Image processing and analysis methods for the Adolescent Brain Cognitive Development Study. *Neuroimage.* 2019;202:116091.
3. Alfaro-Almagro F, Jenkinson M, Bangerter NK, Andersson JLR, Griffanti L, Douaud G, et al. Image processing and Quality Control for the first 10,000 brain imaging datasets from UK Biobank. *Neuroimage.* 2018;166:400–424.
4. Hansen JY, Shafiei G, Markello RD, Smart K, Cox SML, Nørgaard M, et al. Mapping neurotransmitter systems to the structural and functional organization of the human neocortex. *Nat Neurosci.* 2022;25:1569–1581.
5. García-García I, Michaud A, Dadar M, Zeighami Y, Neseliler S, Collins DL, et al. Neuroanatomical differences in obesity: meta-analytic findings and their validation in an independent dataset. *Int J Obes.* 2018;1.
6. Zapparoli L, Devoto F, Giannini G, Zonca S, Gallo F, Paulesu E. Neural structural abnormalities behind altered brain activation in obesity: Evidence from meta-analyses of brain activation and morphometric data. *Neuroimage Clin.* 2022;36:103179.
7. Chen EY, Eickhoff SB, Giovannetti T, Smith DV. Obesity is associated with reduced orbitofrontal cortex volume: A coordinate-based meta-analysis. *Neuroimage Clin.* 2020;28:102420.
8. Herrmann MJ, Tesar AK, Beier J, Berg M, Warrings B. Grey matter alterations in obesity: A meta-analysis of whole-brain studies. *Obesity Reviews.* 2019;20:464–471.
9. Morys F, Dadar M, Dagher A. Association Between Midlife Obesity and Its Metabolic Consequences, Cerebrovascular Disease, and Cognitive Decline. *J Clin Endocrinol Metab.* 2021;106:e4260–e4274.
10. Morys F, Shishikura M, Dagher A. Population-based research in obesity – An overview of neuroimaging studies using big data approach. *Current Opinion in Endocrine and Metabolic Research.* 2022;23:100323.
11. Morys F, Yu E, Shishikura M, Paquola C, Vainik U, Nave G, et al. Neuroanatomical correlates of genetic risk for obesity in children. *Transl Psychiatry.* 2023;13:1.
12. Laurent JS, Watts R, Adise S, Allgaier N, Chaarani B, Garavan H, et al. Associations among Body Mass Index, Cortical Thickness, and Executive Function in Children. *JAMA Pediatr.* 2020;174:170–177.
13. Jiang F, Li G, Ji W, Zhang Y, Wu F, Hu Y, et al. Obesity is associated with decreased gray matter volume in children: a longitudinal study. *Cereb Cortex.* 2023;33:3674–3682.
14. Adise S, Allgaier N, Laurent J, Hahn S, Chaarani B, Owens M, et al. Multimodal brain predictors of current weight and weight gain in children enrolled in the ABCD study ®. *Dev Cogn Neurosci.* 2021;49:100948.

REVIEWERS' COMMENTS:

Reviewer #1 (Remarks to the Author):

The authors responded to all my concerns. I recommend to accept the manuscript for publication.

Reviewer #2 (Remarks to the Author):

The authors provided a response to all the comments raised in the previous letter.
I have no additional comments, and I thank the authors for the clarity of their response.

Reviewer #3 (Remarks to the Author):

The authors have addressed this reviewer's concerns.